# Associations of childhood maltreatment with hypertension in South African women: a cross-sectional study

Kim Anh Nguyen ,[1] Andre Pascal Kengne,[1,2] Naeemah Abrahams,[3,4] Rachel Jewkes,[3,5] Shibe Mhlongo,[3] Nasheeta Peer[1,2]

¹Non-Communicable Diseases Research Unit, South African Medical Research Council, Cape Town and Durban, South Africa
²Department of Medicine, University of Cape Town, Observatory, Western Cape, South Africa
³Gender and Health Research Unit, South African Medical Research Council, Cape Town and Pretoria, South Africa
⁴School of Public Health and Family Medicine, University of Cape Town, Cape Town, South Africa
⁵Office of the Executive Scientist, South African Medical Research Council, Cape Town, South Africa

**Correspondence to**
Kim Anh Nguyen;
Kim.Nguyen@mrc.ac.za

## ABSTRACT

**Objectives** To examine the associations of childhood maltreatment (CM) with hypertension, and the mediating effects of hypertension risk factors on the associations in South African women, using baseline data of the Rape Impact Cohort Evaluation longitudinal study.

**Design** Cross-sectional

**Setting and participants** Self-reported data on CM exposure and its severity in 18–40-year-old women living in KwaZulu-Natal province were assessed. Logistic regression models, adjusted for traditional hypertension risk factors, rape exposure, HIV-infection, other traumatic exposures, depression scores and acute stress reactions (ASR) scores were used to examine the CM–hypertension associations.

**Results** Among 1797 women, 220 (12.2%) had hypertension; CM prevalence was higher in women with hypertension than without hypertension overall (70.9% vs 57.2%) and for each abuse type: sexual abuse (20.9% vs 12.4%), physical abuse (51.8% vs 41.5%), emotional abuse (40% vs 27.6%) and parental neglect (35% vs 25.7%). Exposures to 1–2 types and 3–4 types of CM were 46.4% and 24.5%, respectively, in women with hypertension, and lower in women without (42.9% and 14.3%, respectively). Exposures to any CM (adjusted OR: 1.62; 95% CI: 1.19 to 2.25), sexual abuse (1.64; 95% CI: 1.12 to 2.37), emotional abuse (1.57; 95% CI: 1.16 to 2.13), physical abuse (1.43; 95% CI: 1.07 to 1.92) and parental neglect (1.37; 95% CI: 1.00 to 1.86) were associated with hypertension. Exposures to an increasing number of abuse types and cumulative severity of CM overall (1.13; 95% CI: 1.05 to 1.21) and for each CM type were associated with increased odds of hypertension. Alcohol use, other trauma experienced, depression and ASR partially mediated these associations.

**Conclusion** CM was associated with hypertension; the effects were greater with multiple abuse types and severe abuse, and were partially mediated by alcohol use, depression, ASR and other traumatic exposures. While CM must be prevented, effective mental health interventions to curb the uptake of unhealthy behaviours and the development of hypertension in women exposed to CM are key.

## INTRODUCTION

Hypertension is highly prevalent globally with the greatest burden in sub-Saharan

## STRENGTHS AND LIMITATIONS OF THIS STUDY

⇒ This is among few studies from low-income and middle-income countries to examine the association between childhood maltreatment and hypertension.
⇒ Blood pressures of the study participants were objectively measured during data collection, unlike other relevant studies that used a self-reported measure of hypertension only.
⇒ Four categories of childhood maltreatment, and their frequency and severity, were assessed, which provides a comprehensive understanding of the relationship of childhood maltreatment with hypertension.
⇒ Retrospective self-report of childhood maltreatment experiences could be subject to under reporting.
⇒ Data were collected only in women and might not be applicable to men.

Africa where, at 46%, it is a major contributor to cardiovascular disease morbidity and mortality.[1 2] Together with age, obesity, unhealthy diets, physical inactivity, smoking and alcohol use, psychosocial stressors are among the established drivers of hypertension.[3] Emerging studies have identified exposures to trauma and violence as psychosocial stressors that influence hypertension risk.[4] Childhood traumatic experiences, that is, child abuse is an example of early trauma which may subsequently contribute to the development of hypertension.[5]

Childhood maltreatment (CM) such as sexual abuse, physical abuse, emotional abuse and parental neglect have increasingly been reported in many parts of the world where at least half of the children were exposed to these traumas.[6] The overall prevalence estimate range from 12% reported in Europe, to 50% in Africa and 64% in Asia. The impact of CM is well established with CM-induced traumatic stress contributing to depression, anxiety, post-traumatic stress and maladaptive risky lifestyle behaviours in later life.[7–9] Growing evidence suggests that

CM adversely impacts physical health, resulting in a wide range of disorders including cardiovascular disease risk factors such as hypertension.[7 10 11] It is suggested that exposures to CM may independently activate the hypothalamic–pituitary–adrenal axis and the sympathetic nervous system and promote inflammation, similar to the postulated pathways for other stressors.[4] These physiological changes increase the production of hormones such as catecholamine and glucocorticoid which raise blood pressure (BP) levels. The repeated activation of these systems with multiple or frequent exposures to CM may lead to sustained increases in BP and contribute to the development of hypertension.[4]

Although extensive research has found that chronic psychosocial strain contributes to the development of hypertension overtime,[4] and CM has been considered such a stressor,[5] the current evidence on this relationship is inconsistent. Some studies have demonstrated that CM, particularly sexual abuse and physical abuse, were associated with hypertension.[11] Other studies, however, have shown no such association[12] or only an indirect association via mental health disorders or through the link with obesity.[11] However, these findings were from a few studies conducted mostly in developed countries, with variations in methodologies especially in measurements and definitions of CM as well as the scope of CM investigated.[11–13]

South Africa has high burdens of both hypertension[14] and CM,[15–17] calling for the exploration of an association, if any, between these two factors. Therefore, this study aimed to examine the associations of CM including sexual abuse, physical abuse, emotional abuse and parental neglect with the presence of hypertension in 18–40-year-old South African women, using the baseline data of the Rape Impact Cohort Evaluation (RICE) longitudinal study.[18] Additionally, we explored the mediating effects of potential hypertension risk factors such as body mass index (BMI), glycated haemoglobin (HbA1c), HIV-infection, rape exposure, current smoking, current alcohol consumption, current depressive symptom, acute stress reactions (ASR) and other traumatic exposure on the associations.

## METHODS
### Study design and population
The RICE cohort study was conducted in the Greater Durban Metro area in the KwaZulu-Natal province from October 2014 to March 2020. The study included women who reported being raped (rape exposed) (n=947) recruited from five public health-based postrape care and women who have not ever experienced rape or forced sex (non-rape exposed) (n=852) (controls) recruited from adjacent public health primary healthcare clinics. Both groups of women were likely to be similar because they were recruited from the same geographical area and were of similar ages to ensure comparability.[18 19] The details of the study protocol have been published previously.[18] Eligible participants for both arms were women aged

18–40 years, while those who were more than 14 weeks pregnant, lactating, had cognitive disabilities or severely mentally distressed rape survivors were excluded. Further exclusion among women in the control arm were those who reported lifetime exposure to rape or force sex. Furthermore, the baseline interviews were done within 20 days of the incident rape event among the rape-exposed participants. This timing of the baseline interview was critical for the primary aim of the main study which was to determine the incidence of HIV infection postrape and therefore the baseline documentation of HIV infection and ASR was critical.

Of note, while the RICE study had two arms, in an attempt to achieve the desired aims, the present study used the data from the pooled group of women, which are described in more detail in the statistical analysis.

### Data collection
An electronic data capture system (Bryant) was used for data collection. Trained fieldworkers administered questionnaires, clinical examinations and biochemical assessments. Interview data collected included sociodemographic variables (age, education level, employment status and residence type and area), medical history, lifestyle behaviours (alcohol and tobacco use), exposure to the different childhood exposure types and mental health assessments (depressive symptoms, post-traumatic stress disorder).

### Measurements and definitions
#### Socioemographic and lifestyle characteristics
Education level was categorised as primary (1–7 school years), high-school (8–12 school years) and higher; while residence was classified as formal urban, informal urban, rural and semirural. Current alcohol consumption was defined as consumed at least one drink within a month (one drink equivalent to one can/bottle of beer, cider, cooler/glass of wine/tot of spirit); current smoking as 'currently smoke any tobacco products such as cigarettes, cigars or pipes'.

#### Childhood maltreatment and mental health outcomes
*CM* was assessed using the Childhood Trauma Questionnaire Short Form scale which was adapted and used previously in South Africa.[8 20] Thirteen items measure four abuse types: sexual abuse (four items), emotional abuse (three items), physical abuse (three items) and parental neglect (three items). Participants' exposure to each of these items before 18 years of age was determined. Exposures were evaluated as yes/no for each type of abuse followed by the frequency of exposure (never (score 0), some (score 1) and often (score 2)). Overall CM was defined as exposure to any of the four abuse types and categorised as no exposure (coded 0) and exposure to any abuse (coded 1). Exposure to multiple combined CM types was coded as none (coded 0), 1–2 types (coded 1) and 3–4 types (coded 2). The sum of the frequency of exposure for all CM types described the cumulative

severity. The scores ranged from 0 (no abuse) to 8 (severe or frequent abuse).

The following mental disorders were measured using international scales which were validated in various internationally settings including South Africa.

### Post-traumatic stress disorder (PTSD)/acute stress reactions (ASR)

PTSD are ASRs to a traumatic event that are sustained beyond 1-month post the event. The baseline assessment among the women who experienced rape were done within 20 days of the incident rape, and PTSD due to the rape can therefore not be ascertained and referred to as ASRs.[21] The validated Davidson Trauma self-rating scale was used to measure ASR which included 30 items.[22] Apart from being validated internationally, the scale has been used in other contexts in South Africa.[23 24] Participants were asked how much each item/symptom affected them over the past week, with responses evaluated on a 5-point Likert scale: 'not at all' (0), 'only once' (1), '2–3 times' (2), '4–6 times' (3) and 'all the time' (4). Scores for all items were summed for an ASR score.

### Current depressive symptoms

These were measured using the 20-item Center for Epidemiologic Studies Depression (CESD) self-report measure,[25] which has been validated in international settings including South Africa.[24 26] The CESD scale was designed to assess how often a person experienced depressive symptoms within the past week. The response was categorised into a 4-point Likert scale ranging from 'rarely or none of the time' (0) to 'most or all of the times' (3). The summed score provided the overall CESD score with higher scores indicating more severe depressive symptoms.

Exposures to other prior traumatic events were assessed using an adapted Life Event Checklist.[23 24 27] The checklist includes 10 items of traumatic events namely imprisonment, civil unrest/war, serious injury, being close to death, witnessing a murder of family or friend, unnatural death of family or friend, witnessing the murder of stranger, torture, robbed or carjacked at gun or knife point and kidnapping. The yes (score 1)/no (score 0) scores for all the items were totalled to create a continuous variable of 'other traumatic exposures'. A higher score was indicative of more lifetime traumatic events or exposures.

### Clinical parameters

Anthropometric measures were done using standardised methods. Height was measured to the nearest millimetre with the participant in an upright position and barefooted. Weight was taken to the nearest 0.1 kg with the participants in light clothing and without shoes. BMI was calculated as weight in kilograms divided by height in metres squared, and BMI ≥25 kg/m$^2$ was used to define overweight or obese. Waist circumference (WC) was recorded to the nearest millimetre at the level that is midway between the upper border of the hip bone and the lower border of the lowest rib. Elevated WC was defined as WC >80 cm.[28]

BP was measured using a digital BP monitor (Omron, M6 Comfort, the Netherlands) after the participant was seated in a resting position for at least 5 min; three measurements were taken 3 min apart and the average of the second and third readings were used in the analysis. Hypertension was defined as systolic BP ≥140 mm Hg or diastolic BP ≥90 mm Hg or self-reported previous diagnosis of hypertension.[3]

### Biochemical parameters

Biochemical measurements were done at an accredited pathology laboratory (Global Laboratory, Durban, South Africa). Serum lipids were analysed by enzymatic colorimetric method; HbA1c was measured using high-performance liquid chromatography (VARIANT II TURBO. EDTA tubes) following the National Glycathaemoglobin Standardisation Programme. Dyslipidaemia was based on the South African national lipid guidelines and defined as follows: total cholesterol>5 mmol/L, low-density lipoprotein cholesterol>3 mmol/L, triglycerides>1.5 mmol/L and high-density lipoprotein cholesterol<1.2 mmol/L.[29] Dysglycaemia was defined as HbA1c≥5.7% and or previously diagnosed with diabetes.[30]

### Statistical analyses

Data were analysed using R statistical software V.3.6.0 (26 April 2019). Since this study aimed to determine the associations of CM with adult hypertension in both rape-exposed and non-rape-exposed women, and not to determine and/or compare the prevalence of CM and outcome hypertension between the two groups of women; this analysis used the baseline data from the pooled group of women and not just the women who experienced rape. Exploratory analyses revealed no significant interaction effect of 'rape exposure' on the association of CM with hypertension status, motivating the pooled analyses of both groups (women who experienced rape and women who did not) to maximise the statistical power. This was done using logistic regression models where the interaction term (CM*rape-exposure) was incorporated in the same model for the effects of CM on outcome hypertension.

The baseline characteristics were summarised as means (SD) or medians (25th–75th percentiles) for continuous variables, and as count (percentages) for categorical variables in the overall sample and by hypertension status. Comparison of baseline characteristics by hypertension was using $\chi^2$ tests, fisher-exact test, t-tests or Kruskal-Wallis tests for non-parametric data where appropriate. The relations of CM with hypertension were explored across (1) each of the four CM categories, (2) multiple combined CM categories and (3) cumulative frequency and severity of CM. The associations were explored using multiple logistic regression analyses with adjustments for potential confounders and mediators. The potential confounders were identified as those variables which

were likely correlated with CM and causally linked to the development of hypertension and included age, the recent rape exposure, education level, employment status and residence type.

The potential mediators were based on the existing literature and included BMI, HbA1c, smoking, alcohol-consumption, HIV infection, depressive symptoms, ASR and other traumatic experiences. They were postulated to be involved in the pathways linking CM with hypertension. HIV infection was considered as a potential mediator because previous studies in South Africa have shown CM to be associated with HIV-related risk behaviour,[16] and living with HIV infection was associated higher risk for hypertension.[31] On rare occasions, HIV infection could have been acquired before CM exposure, but the number would likely be too few to bias the mediation analyses. Other traumatic experience was positioned as a potential mediator based on the local literature.[32] Although the exact order of occurrence between CM and other traumatic exposure could not be specified in the current study, CM, by definition, typically occurred only in childhood. Some traumatic events could have happened at any time during the lifespan, and have been conventionally measured through scores that capture the lifetime experience; which were used in the current study. Lifetime experience of these other traumatic events could have been the consequence of CM or not; but CM would very unlikely be the consequence of other traumatic exposure.

Variables not found to be associated with hypertension in the bivariate analysis (p>0.05) were subsequently excluded from the regression models. The latter included education level, employment status and residence type. Rape exposure variable remained in the models as adjustment variable because it forms part of study design.

The mediation effects of possible mediators adjusted for potential confounders (age and rape-exposure) on the association of CM (any CM, multiple combined CM types and the cumulative severity score) with hypertension were explored using simple mediation analysis.[33] Each mediator was analysed separately in individual mediation models, which comprised the hypertension outcome variable, a predictor (CM exposure) and a mediator. *Laavan* package (http://CRAN.R-project.org/package=lavaan) was used to compute the indirect, direct and total effects estimates, and proportions of mediation effects (indirect effect/total effect), with the significance of the mediation effect tested via bootstrap methods, based on 5000 replications. Estimates are reported as standardised beta coefficients which were derived based on diagonal-weighted least square method, but also as OR and 95% CIs. It is of note, however, that the interpretation of OR from probit models (implemented for mediation analysis in *Laavan*) is not straightforward. We further explored the mediating effects using multiple mediation model with the same adjustment variables. A p-value<0.05 was considered a statistically significant mediation effect. Full or complete mediation is present when the total and indirect effects are significant, while the direct effect is

non-significant. Partial mediation occurs when the total and indirect effects are significant, and the direct effect remains significant.

Furthermore, we have replicated both simple and multiple mediation analyses using 'mediation' package (http://CRAN.R-project.org/package=mediation), and presented the findings as secondary analyses.[34]

## Patient and public involvement

In this study, the research questions and outcomes were not informed by patients' priorities, experiences and preferences. Patient and the public were not involved in the design or conduct or reporting or dissemination plan.

## RESULTS

A total of 1799 women were recruited into the study. Two participants had missing data on CM and were excluded. Therefore, the final sample consisted of 1797 participants. Of these, 220 (12.2%) had prevalent hypertension. The characteristics of the participants are presented in table 1. The overall median age was 24 years; participants with hypertension were older compared with their counterparts without hypertension (26 years vs 24 years, p<0.001). Compared with those without hypertension, women with hypertension were more likely to be HIV-positive and to currently drink alcohol; they had higher scores for depression, ASR and were more likely to have experienced other traumatic events in their lives (all p≤0.05). Participants with hypertension had higher mean WC and BMI levels and dysglycaemia, but their lipid profiles were similar to those without hypertension. There was no difference in terms of education, residence, current rape exposure or smoking status between the participants with and without hypertension (table 1).

## Prevalence of CM types by hypertensive status

The distribution of CM types overall and by hypertension status is presented in table 2. The prevalence of CM was 58.9% in overall sample, with physical abuse being the most frequent (42.7%). This was followed by emotional abuse and parental neglect at 29% each, while exposure to sexual abuse was 13.4%. The prevalence of all four CM types was significantly higher in participants with, than without, hypertension. The prevalence of CM was 70.9% in women with hypertension versus 57.2% in those without hypertension (p<0.001). More than half of the women with hypertension (51.8%) had experienced physical abuse compared with 41.5% of women without hypertension (p=0.004). A fifth (20.9%) versus 12.4% of women with hypertension compared with without hypertension were exposed to sexual abuse in childhood. Exposures to 1–2 types and 3–4 types of CM were 46.4% and 24.5%, respectively, in women with hypertension, and lower in women without (42.9% and 14.3%, respectively) (p<0.001). Similar patterns of higher prevalence rates in women with hypertension compared with without hypertension were observed for the severity and frequency of

**Table 1** Sociodemographic characteristics and cardiovascular disease risk factors presented by hypertension status

| Variables | Overall (n=1797) | Hypertension (n=220) | No hypertension (n=1577) | P value |
|---|---|---|---|---|
| Age in years, median (P25–P75) | 24 (21–29) | 26 (23–30) | 24 (21–28) | <0.001 |
| Women who experienced rape | 47.4 (45–50) | 53.6 (46.8–60.4) | 46.4 (39.6–53.2) | 0.054 |
| Years schooling, % (95% CI) | | | | 0.415 |
| 1–7 | 2.8 (1.4 to 4.2) | 3.6 (0.4 to 7.6) | 2.7 (1.2 to 4.1) | |
| 8–12 | 89 (87.6 to 90.4) | 90 (86.8 to 93.9) | 88.8 (87.4 to 90.3) | |
| >12 | 8.2 (6.9 to 9.6) | 6.4 (3.2 to 10.3) | 8.5 (7 to 10) | |
| Employment status, % (95% CI) | | | | |
| Employed, % (95% CI) | 20.5 (18.6 to 22.4) | 24.1 (18.6 to 30.3) | 20 (18 to 22) | 0.184 |
| Residence, % (95% CI) | n=1774 | n=218 | n=1556 | 0.790 |
| Formal urban | 72 (70 to 74) | 74 (68.3 to 79.6) | 71.7 (69.5 to 74) | |
| Informal urban | 16.6 (14.5 to 18.7) | 15 (9.6 to 20.9) | 16.8 (14.6 to 19) | |
| Rural/semirural | 11.4 (9.4 to 13.6) | 11 (5.5 to 16.7) | 11.5 (9.3 to 13.8) | |
| HIV-positive, % (95% CI) | 43.3 (40.9 to 45.6) | 52.7 (45.9 to 59.5) | 42 (39.5 to 44.4) | 0.003 |
| Lifetime trauma event exposure | | | | |
| Other traumatic exposures score, median (P25–P75) | 2 (1–3) | 2 (1–4) | 2 (1–3) | <0.001 |
| Health behaviours, % (95% CI) | | | | |
| Current smoker | 12.5 (11 to 14) | 13.6 (9.4 to 18.9) | 12.3 (10.7 to 14) | 0.651 |
| Current alcohol consumption | 53.1 (50.7 to 55.4) | 61.8 (55 to 68.2) | 51.9 (49.4 to 54.4) | 0.006 |
| Mental health disorders | | | | |
| Depression score | 21 (11–36) | 20 (11–36) | 28 (14–38) | <0.001 |
| ASR score, median (P25–P75) | 16 (2–37) | 22 (5–41) | 15 (1–37) | 0.001 |
| Cardiometabolic health | | | | |
| *Mean (SD) or median (P25–P75)* | | | | |
| Systolic blood pressure, mm Hg | 105 (8.5) | 108 (11.2) | 104 (7.9) | <0.001 |
| Diastolic blood pressure, mm Hg | 70 (7.6) | 73 (10.3) | 70 (7.1) | <0.001 |
| Waist circumference, cm | 84 (13.9) | 88 (15) | 84 (13.7) | <0.001 |
| Body mass index | 27 (6.6) | 29 (8.3) | 27 (6.3) | <0.001 |
| | **n=1614** | **n=169** | **n=1445** | |
| Total cholesterol, mmol/L | 3.8 (0.82) | 4.0 (0.84) | 3.8 (0.81) | 0.052 |
| Low-density lipoprotein cholesterol (LDL-C), mmol/L | 2.2 (0.73) | 2.3 (0.74) | 2.2 (0.72) | 0.385 |
| High-density lipoprotein cholesterol (HDL-C), mmol/L | 1.2 (0.33) | 1.25 (0.37) | 1.2 (0.33) | 0.128 |
| Triglycerides, mmol/L | 0.72 (0.52–0.98) | 0.74 (0.55–1.01) | 0.72 (0.52–0.98) | 0.402 |
| Glycated haemoglobin (HbA1c), % | 5.3 (0.6) | 5.4 (0.7) | 5.3 (0.5) | 0.169 |
| *Prevalence, % (95% CI)* | | | | |

Continued

**Table 1** Continued

| Variables | Overall (n=1797) | Hypertension (n=220) | No hypertension (n=1577) | P value |
|---|---|---|---|---|
| Waist circumference ≥80 cm | 57.6 (55.3 to 59.9) | 64.1 (57.4 to 70.4) | 56.7 (54.3 to 59.2) | 0.046 |
| Body mass index ≥25 kg/m$^2$ | 54.4 (52.1 to 56.7) | 60.4 (53.7 to 66.9) | 53.6 (51.1 to 56.1) | 0.065 |
| Total cholesterol >5 mmol/L | 7.5 (6.3 to 8.9) | 8.9 (5.1 to 14.2) | 7.3 (6 to 8.8) | 0.572 |
| LDL-C>3 mmol/L | 13.5 (11.9 to 15.3) | 16 (10.8 to 22.4) | 13.2 (11.5 to 15.1) | 0.382 |
| HDL-C<1.2 mmol/L | 53.6 (51.1 to 56) | 49.7 (41.9 to 57.5) | 54 (51.4 to 56.6) | 0.322 |
| Triglycerides >1.5 mmol/L | 7.2 (6 to 8.6) | 7.1 (3.7 to 12.1) | 7.3 (6 to 8.7) | 0.999 |
| HbA1c ≥5.7% | 21.1 (19.2 to 23.1) | 30 (24 to 36.5) | 19.9 (17.9 to 21.9) | 0.001 |
| History of known diabetes | 1.4 (0.9 to 2) | 2.7 (1 to 5.8) | 1.2 (0.7 to 1.9) | 0.112 |
| History of known hypertension | 11.6 (10.1 to 13.1) | 94.5 (90.7 to 97.2) | NA | NA |

Hypertension defined as follows: systolic blood pressure ≥140 mm Hg and/or diastolic blood pressure ≥90 mm Hg or previously diagnosed with hypertension; current smoker: currently smoke any tobacco products such as cigarettes, cigars or pipes; current alcohol consumption, consumed at least one drink in an occasion within a month, respectively (one drink is equivalent to one can/bottle of beer, cider, cooler/glass of wine/tot of spirit); severe depression: total scores ≥16 for depressive symptoms during the past week using Centre for Epidemiologic Studies Depression Scale (CES-D score); ASR, acute stress reactions, was measured using Davidson Trauma Scale. P-values for $\chi^2$ tests, fisher-exact test, t-tests or Kruskal-Wallis tests for non-parametric data where appropriate.

abuse experienced (all p≤0.003). There was no evidence of a statistically significant interaction effect of rape exposure on the distribution of CM by hypertension status as shown in table 3 (all interaction p≥0.061).

### Associations of CM exposure with prevalent hypertension

The ORs for the relationship of CM with hypertension adjusted for age, BMI, current smoking, current alcohol use, recent rape exposure, HIV infection, other traumatic exposure, depression scores, ASR scores and dysglycaemia are presented in table 3. There was a positive association of exposure to any CM (adjusted OR (AOR): 1.62; 95% CI: 1.19 to 2.25), sexual abuse (AOR: 1.64; 95% CI: 1.12 to 2.37), physical abuse (AOR: 1.43; 95% CI: 1.07 to 1.92) and emotional abuse (AOR: 1.57; 95% CI: 1.16 to 2.13) with prevalent hypertension. The association of exposure to parental neglect (AOR: 1.37; 95% CI: 1.00 to 1.86) was borderline. Increasing number of maltreatment types as well as cumulative severity/frequency of maltreatment overall or by maltreatment type were associated with increased odds of hypertension (table 3).

The results of the mediation analysis showed that the associations of CM (any CM, multiple combined CM types and the cumulative severity score) with hypertension were partially mediated by current alcohol consumption (all p≤0.043 for the mediation effects), depressive symptom score (all p≤0.022 for the mediation effects) and experiencing other traumatic events in life (all p<0.001 for the mediation effects) (table 4 and online supplemental table S1). ASR score partially mediated the association of any CM (p=0.024 for the mediation effect) and multiple CM types (p=0.045 for the mediation effect) but cumulative severity abuse score (p=0.066 for the mediation effect). BMI, HbA1c, HIV status and smoking habit had no significant mediation effects on the association between CM

exposure and hypertension (all p≥0.144 for the mediation effects).

When the multiple mediation analysis model adjusted for age and rape exposure was applied (table 5 and online supplemental table S2), the direct effects of any CM (p=0.084 for the direct effect) were completely mediated by current alcohol consumption (p=0.034 for mediation effect), depressive symptom score (p=0.040) and experiencing other traumatic events (p<0.001). The direct effects of multiple CM types and the cumulative severity abuse score on hypertension remained significant, and the associations were partially mediated by current alcohol consumption and experiencing other traumatic events (figure 1).

Results from secondary analyses of both simple and multiple mediation models showed similar patterns (online supplemental tables S3 and S4). In simple mediation models, the associations of any CM and multiple combined CM types with hypertension were partially mediated by current alcohol consumption (both ps≤0.036 for the average causal mediation effects (ACMEs), depressive symptom score (both ps≤0.026 for the ACMEs) and lifetime experiences of other traumatic events (both ps<0.001 for the ACMEs) (online supplemental table S3). The associations of the cumulative severity score with hypertension were partially mediated by lifetime experiences of other traumatic events (p<0.001 for the ACME). ASR, BMI, HbA1c, HIV status and smoking habit had no significant mediation effects on the association between CM exposure and hypertension. In multiple mediation models, the association of any CM with hypertension was partially mediated by depressive symptom score and lifetime experiences of other traumatic events (online supplemental table S4). The associations of multiple

**Table 2** Prevalence of childhood maltreatment types overall and by hypertension status

| Number (%) | Overall (n=1797) | | | Hypertension (n=220) | | | No hypertension (n=1577) | | | P value |
|---|---|---|---|---|---|---|---|---|---|---|
| | n | % | 95% CI | n | % | 95% CI | n | % | 95% CI | |
| **Ever exposed to childhood maltreatment** | | | | | | | | | | |
| Sexual abuse | 241 | 13.4 | 11.9 to 15.1 | 46 | 20.9 | 15.7 to 26.9 | 195 | 12.4 | 10.8 to 14.1 | <0.001 |
| Physical abuse | 768 | 42.7 | 40.4 to 45.1 | 114 | 51.8 | 41.4 to 54.9 | 654 | 41.5 | 39 to 43.9 | 0.004 |
| Emotional abuse | 523 | 29.1 | 27 to 31.3 | 88 | 40.0 | 33.4 to 46.8 | 435 | 27.6 | 25.4 to 29.9 | <0.001 |
| Parental neglect | 482 | 26.8 | 24.8 to 28.9 | 77 | 35.0 | 28.7 to 41.7 | 405 | 25.7 | 23.5 to 27.9 | 0.004 |
| Any childhood maltreatment | 1058 | 58.9 | 56.5 to 61.2 | 156 | 70.9 | 64.4 to 76.8 | 902 | 57.2 | 54.7 to 59.6 | <0.001 |
| **Exposure to multiple types of childhood maltreatment** | | | | | | | | | | <0.001 |
| No abuse | 739 | 41.1 | 38.6 to 43.6 | 64 | 29.1 | 22.3 to 36.4 | 675 | 42.8 | 40.1 to 45.5 | |
| 1–2 types | 778 | 43.3 | 40.8 to 45.8 | 102 | 46.4 | 39.5 to 53.7 | 676 | 42.9 | 40.2 to 45.5 | |
| 3–4 types | 280 | 15.6 | 13.1 to 18.1 | 54 | 24.5 | 17.7 to 31.8 | 226 | 14.3 | 11.7 to 17 | |
| **Combined frequency and severity of childhood maltreatment** | | | | | | | | | | |
| **Sexual abuse** | | | | | | | | | | <0.001 |
| Never | 1556 | 86.6 | 85.1 to 88.1 | 174 | 79.1 | 74.1 to 84.2 | 1382 | 87.6 | 86.1 to 89.2 | |
| Some | 156 | 8.7 | 7.2 to 10.2 | 25 | 11.4 | 6.4 to 16.5 | 131 | 8.3 | 6.8 to 9.9 | |
| Often | 85 | 4.7 | 3.3 to 6.3 | 21 | 9.5 | 4.5 to 14.7 | 64 | 4.1 | 2.5 to 5.6 | |
| **Physical abuse** | | | | | | | | | | 0.002 |
| Never | 1029 | 57.3 | 54.9 to 59.7 | 106 | 48.2 | 41.4 to 55.4 | 923 | 58.5 | 56.1 to 61.1 | |
| Some | 378 | 21.0 | 18.7 to 23.5 | 48 | 21.8 | 15 to 29 | 330 | 20.9 | 18.5 to 23.5 | |
| Often | 390 | 21.7 | 19.4 to 24.1 | 66 | 30.0 | 23.2 to 37.2 | 324 | 20.5 | 18.1 to 23.1 | |
| **Emotional abuse** | | | | | | | | | | <0.001 |
| Never | 1274 | 70.9 | 68.8 to 73 | 132 | 60.0 | 53.6 to 66.8 | 1142 | 72.4 | 70.3 to 74.6 | |
| Some | 290 | 16.1 | 14.1 to 18.3 | 44 | 20.0 | 13.6 to 26.7 | 246 | 15.6 | 13.4 to 17.8 | |
| Often | 233 | 13.0 | 10.9 to 15.1 | 44 | 20.0 | 13.6 to 26.7 | 189 | 12.0 | 9.8 to 14.2 | |
| **Parental neglect** | | | | | | | | | | 0.003 |
| Never | 1315 | 73.2 | 71.2 to 75.2 | 143 | 65.0 | 59.1 to 71.6 | 1172 | 74.3 | 72.2 to 76.5 | |
| Some | 328 | 18.3 | 16.2 to 20.3 | 47 | 21.4 | 15.4 to 28 | 281 | 17.8 | 15.7 to 20 | |
| Often | 154 | 8.6 | 6.6 to 10.6 | 30 | 13.6 | 7.7 to 20.3 | 124 | 7.9 | 5.8 to 10 | |

Frequency and severity of childhood maltreatment defined as: Some, affirmative response of 'sometimes' to one item only in a specific maltreatment category; Often, affirmative response of 'sometime' to >1 item, or response of 'often' or 'very often' to at least one item in a specific maltreatment category. P-values for $\chi^2$ tests or fisher-exact tests where appropriate.

**Table 3** Logistic regression analyses for the associations of childhood maltreatment with hypertension

| Childhood maltreatment (CM) variable | Sample size | Hypertension n (%) | Unadjusted OR | P value | P interaction (CM*RE) | Adjusted OR | P value |
|---|---|---|---|---|---|---|---|
| Ever exposed to CM | | | | | | | |
| Any CM | | | | | 0.083 | | |
| No | 739 | 64 (8.7) | 1.00 | | | 1.00 | |
| Yes | 1058 | 156 (14.7) | 1.82 (1.35–2.50) | <0.001 | | 1.62 (1.19–2.25) | 0.002 |
| Sexual abuse | | | | | 0.081 | | |
| No | 1556 | 174 (11.2) | 1.00 | | | 1.00 | |
| Yes | 241 | 46 (19.1) | 1.87 (1.30–2.66) | <0.001 | | 1.64 (1.12–2.37) | 0.009 |
| Physical abuse | | | | | 0.468 | | |
| No | 1029 | 106 (10.3) | 1.00 | | | 1.00 | |
| Yes | 768 | 114 (14.8) | 1.52 (1.14–2.02) | 0.004 | | 1.43 (1.07–1.92) | 0.016 |
| Emotional abuse | | | | | 0.201 | | |
| No | 1274 | 132 (10.4) | 1.00 | | | 1.00 | |
| Yes | 523 | 88 (16.8) | 1.75 (1.30–2.34) | <0.001 | | 1.57 (1.16–2.13) | 0.003 |
| Parental neglect | | | | | 0.557 | | |
| No | 1315 | 143 (10.9) | 1.00 | | | 1.00 | |
| Yes | 482 | 77 (16.0) | 1.56 (1.15–2.09) | 0.004 | | 1.37 (1.00–1.86) | 0.049 |
| Exposure to multiple types of CM | | | | | 0.101 | | |
| No abuse | 739 | 64 (8.7) | 1.00 | | | 1.00 | |
| 1–2 abuse types | 778 | 102 (13.1) | 1.59 (1.15–2.22) | 0.005 | | 1.41 (1.01–1.99) | 0.046 |
| 3–4 abuse types | 280 | 54 (19.3) | 2.52 (1.70–3.72) | <0.001 | | 1.81 (1.18–2.76) | 0.006 |
| Frequency and severity of CM | | | | | | | |
| Cumulative maltreatment (range 0–8) | 1797 | 220 (12.2) | 1.19 (1.12–1.28) | <0.001 | 0.061 | 1.13 (1.05–1.21) | 0.002 |
| Sexual abuse | | | | | 0.081 | | |
| Never | 1556 | 174 (11.2) | 1.00 | | | 1.00 | |
| Some | 156 | 25 (16.0) | 1.52 (0.94–2.35) | 0.073 | | 1.39 (0.86–2.19) | 0.164 |
| Often | 85 | 21 (24.7) | 2.61 (1.52–4.30) | <0.001 | | 2.12 (1.21–3.60) | 0.007 |
| Physical abuse | | | | | 0.378 | | |
| Never | 1029 | 106 (10.3) | 1.00 | | | 1.00 | |
| Some | 378 | 48 (12.7) | 1.27 (0.87–1.81) | 0.202 | | 1.24 (0.85–1.79) | 0.246 |
| Often | 390 | 66 (16.9) | 1.77 (1.27–2.47) | <0.001 | | 1.62 (1.14–2.27) | 0.006 |
| Emotional abuse | | | | | 0.116 | | |

Continued

**Table 3** Continued

| Childhood maltreatment (CM) variable | Sample size | Hypertension n (%) | Unadjusted OR | P value | P interaction (CM*RE) | Adjusted OR | P value |
|---|---|---|---|---|---|---|---|
| Never | 1274 | 132 (10.4) | 1.00 | | | 1.00 | |
| Some | 290 | 44 (15.2) | 1.55 (1.06–2.22) | 0.020 | | 1.44 (0.97–2.08) | 0.061 |
| Often | 233 | 44 (18.9) | 2.01 (1.37–2.91) | <0.001 | 0.139 | 1.75 (1.17–2.57) | 0.005 |
| Parental neglect | | | | | | | |
| Never | 1315 | 143 (10.9) | 1.00 | | | 1.00 | |
| Some | 328 | 47 (14.3) | 1.37 (0.95–1.94) | 0.081 | | 1.21 (0.84–1.73) | 0.302 |
| Often | 154 | 30 (19.5) | 1.98 (1.26–3.03) | 0.002 | | 1.74 (1.09–2.71) | 0.016 |

Logistic regression models adjusted for age, BMI, recent rape exposure, current smoking; current alcohol use, HIV infection, dysglycaemia (HbA1c≥5.7% and or previously diagnosed with diabetes),other traumatic exposures scores, depression scores and acute stress reactions (ASR) scores. Current smoking: currently smoke any tobacco products such as cigarettes, cigars or pipes; current alcohol consumption: consumed at least one drink in an occasion within a month, respectively (one drink is equivalent to one can/bottle of beer, cider, cooler/glass of wine/tot of spirit); depression scores: depressive symptoms during the past week using Center for Epidemiologic Studies Depression Scale (CES-D score); ASR (scores): acute stress reactions measured using Davidson Trauma Scale, a validated self-rating scale. Cumulative maltreatment: ranged from 0 (no maltreated in any type) to 8 (often/severely maltreated in all 4 types).
BMI, body mass index; CM, childhood maltreatment; RE, rape exposure;;

combined CM types and the cumulative severity score with hypertension were partially mediated by lifetime experiences of other traumatic events. The results of sensitivity analyses suggest that the point estimates of the ACME are rather sensitive to the violation of the sequential ignorability assumption (online supplemental figures S1 and S2).

## DISCUSSION

In this study among South African adult women attending specific healthcare services, CM was widespread and associated with hypertension. To the best of our knowledge, this is among very few studies conducted in low-income and middle-income countries, which describes significant associations between CM and hypertension in women after adjusting for traditional hypertension risk factors, other traumatic exposures and poor mental health. The study showed exposure to sexual, physical, emotional abuse, multiple (3–4) abuse types and frequent and severe exposure were significantly associated with hypertension. Furthermore, it showed that the associations were potentially mediated by alcohol use, other traumatic exposure, depression and ASR.

These findings accord with the literature from high-income countries; frequent and severe exposure to physical, emotional and sexual abuse and parental neglect in childhood and exposure to multiple abuse types compared with lesser abuse exposures may have greater negative health consequences,[35] including hypertension.[36 37] The latter includes two longitudinal studies from the USA. Among 394 men and women followed for 23 years, those with multiple CM exposures showed a greater increase in adult BP levels compared with those without childhood traumatic experiences.[36] The Nurses' Health Study II showed that severe childhood physical and/or sexual abuse was associated with an increased risk of hypertension in adulthood independent of race, oral contraceptive use, BMI, smoking, alcohol use, exercise and child somatogram score.[37]

However, these two studies, unlike the current study, did not investigate the effects of mental health issues such as depression and ASR, on the relationship between CM and hypertension. Notably, the current study found that the associations of CM with hypertension remained after adjusting for poor mental health-related factors for all abuse types. Current depressive symptom, ASR and other prior traumatic events mediated the relationships between CM and hypertension in this study. The additive influences of these psychological factors likely demonstrate their role in the development of hypertension. While a detailed exploration of these associations is beyond the scope of this paper, the World Mental Health Survey reported similar findings. The presence of three or more childhood adversities (abuse, neglect and other traumatic experiences) was modestly associated with self-reported hypertension, with the association not fully explained by early-onset or current depression-anxiety.[38]

**Table 4** Results of simple mediation analysis for the effects of childhood maltreatment on adult prevalent hypertension adjusted for age and rape exposure in a sample of South African women (n=1797)

| Confounder/mediator | Total effect | | | Direct effect | | | Indirect effect | | | Proportions of mediation effects |
|---|---|---|---|---|---|---|---|---|---|---|
| | Estimates | 95% CI | P value | Estimates | 95% CI | P value | Estimates | 95% CI | P value | % (95% CI) |
| Any CM | 0.312 | 0.151 to 0484 | <0.001 | | | | | | | |
| BMI | | | | 0.301 | 0.140 to 0.463 | <0.001 | 0.011 | −0.004 to 0.026 | 0.164 | 3.5 (−1.6 to 11) |
| HbA1c | | | | 0.314 | 0.151 to 0.473 | <0.001 | −0.002 | −0.015 to 0.011 | 0.750 | −0.6 (−5.5 to 1.6) |
| HIV-positive | | | | 0.310 | 0.148 to 0.471 | <0.001 | 0.002 | −0.014 to 0.018 | 0.823 | 0.9 (2.9 to 6.3) |
| Current smoking | | | | 0.308 | 0.144 to 0.468 | <0.001 | 0.003 | −0.029 to 0.034 | 0.872 | 0.5 (−3.8 to 6.8) |
| Current alcohol | | | | 0.282 | 0.119 to 0.444 | <0.001 | 0.029 | 0.001 to 0.058 | 0.049 | 5.6 (1.2 to 16.7) |
| Depressive symptom score | | | | 0.282 | 0.122 to 0.443 | <0.001 | 0.029 | 0.004 to 0.053 | 0.022 | 12.5 (3.6 to 29.5) |
| ASR score | | | | 0.287 | 0.124 to 0.449 | <0.001 | 0.024 | 0.008 to 0.055 | 0.045 | 10.5 (1.5 to 29) |
| Other traumatic exposures score | | | | 0.232 | 0.067 to 0.397 | 0.006 | 0.081 | 0.045 to 0.118 | <0.001 | 27.2 (16.9 to 68.6) |
| Multiple CM types | 0.243 | 0.138 to 0.349 | <0.001 | | | | | | | |
| BMI | | | | 0.236 | 0.130 to 0.342 | <0.001 | 0.008 | −0.003 to 0.018 | 0.157 | 2.1 (1.5 to 8.4) |
| HbA1c | | | | 0.244 | 0.138 to 0.349 | <0.001 | −0.001 | −0.009 to 0.007 | 0.777 | 0.1 (−3.3 to 2.2) |
| HIV-positive | | | | 0.238 | 0.132 to 0.344 | <0.001 | 0.005 | −0.006 to 0.017 | 0.374 | 1.8 (0.9 to 6.8) |
| Current smoking | | | | 0.240 | 0.132 to 0.348 | <0.001 | 0.001 | −0.026 to 0.027 | 0.959 | 0.1 (−4.9 to 5.9) |
| Current alcohol | | | | 0.223 | 0.116 to 0.330 | <0.001 | 0.020 | 0.001 to 0.040 | 0.046 | 5.6 (0.8 to 12.4) |
| Depressive symptom score | | | | 0.220 | 0.113 to 0.326 | <0.001 | 0.022 | 0.001 to 0.043 | 0.037 | 11 (3 to 26) |
| ASR score | | | | 0.226 | 0.119 to 0.333 | <0.001 | 0.016 | 0.001 to 0.039 | 0.051 | 9.2 (1.0 to 25.3) |
| Other traumatic exposures score | | | | 0.178 | 0.069 to 0.287 | 0.001 | 0.066 | 0.057 to 0.095 | <0.001 | 30.9 (15 to 60) |
| Cumulative CM score | 0.094 | 0.055 to 0.132 | <0.001 | | | | | | | |
| BMI | | | | 0.092 | 0.053 to 0.130 | <0.001 | 0.002 | −0.002 to 0.005 | 0.363 | 0.3 (−3.3 to 6.5) |
| HbA1c | | | | 0.094 | 0.056 to 0.133 | <0.001 | −0.001 | −0.004 to 0.001 | 0.347 | −0.4 (−4.2 to 1.2) |
| HIV-positive | | | | 0.092 | 0.054 to 0.131 | <0.001 | 0.001 | −0.004 to 0.005 | 0.704 | 1.1 (−1.4 to 5.5) |
| Current smoking | | | | 0.092 | 0.053 to 0.132 | <0.001 | 0.000 | −0.010 to 0.011 | 0.994 | −0.4 (−5.1 to 6) |
| Current alcohol | | | | 0.086 | 0.048 to 0.126 | <0.001 | 0.007 | 0.001 to 0.014 | 0.049 | 4.5 (0.6 to 12) |
| Depressive symptom score | | | | 0.084 | 0.045 to 0.123 | <0.001 | 0.008 | 0.001 to 0.017 | 0.047 | 11.1 (2.1 to 24.8) |

Continued

**Table 4** Continued

| Confounder/ mediator | Total effect | | | Direct effect | | | Indirect effect | | | Proportions of mediation effects |
|---|---|---|---|---|---|---|---|---|---|---|
| | Estimates | 95% CI | P value | Estimates | 95% CI | P value | Estimates | 95% CI | P value | % (95% CI) |
| ASR score | | | | 0.087 | 0.047 to 0.126 | <0.001 | 0.006 | 0.000 to 0.016 | 0.193 | 8.6 (0.8 to 22.4) |
| Other traumatic exposures score | | | | 0.067 | 0.027 to 0.107 | 0.001 | 0.027 | 0.015 to 0.038 | <0.001 | 31.5 (16.5 to 65.4) |

Data are standardised beta coefficient; current smoking: currently smoke any tobacco products such as cigarettes, cigars or pipes; current alcohol consumption: consumed at least one drink in an occasion within a month, respectively (one drink is equivalent to one can/bottle of beer, cider, cooler/glass of wine/tot of spirit); depressive symptoms (scores): depressive symptoms during the past week using Centre for Epidemiologic Studies Depression Scale (CES-D score); ASR (scores): acute stress reactions measured using Davidson Trauma Scale, a validated self-rating scale. Cumulative abuse (score): ranged from 0 (no abused in any type) to 8 (often/severely abused in all four types).
BMI, body mass index; CM, childhood maltreatment.

In addition, a survey in Brazil found childhood physical abuse and family violence were associated with a greater likelihood of adult hypertension; the association was attenuated by the onset of depression in adulthood.[13] Although CM was associated with hypertension independent of current mental health disorders in the current study, it is important to investigate whether the association is reversible. Such understanding will help to design interventions to prevent the development of hypertension in the CM victims.

While exposures to any CM, sexual abuse and emotional abuse were significantly associated with hypertension, there was weaker effect of physical abuse or parental neglect with hypertension in the present study. More research, particularly from longitudinal studies, is required for a deeper understanding of these associations and to examine the differential impact of the various types of CM on the development of hypertension. Similar to these findings, a prospective childhood study in the USA reported no significant differences in the associations of physical abuse and neglect with the development of hypertension after 30 years of follow-up.[39] However, unlike the current study, they also found no association between childhood sexual abuse and hypertension. The study controlled for age, gender and race but did not assess the severity of CM.

Another study that examined childhood physical and verbal abuse and neglect using a short-form family environment questionnaire found an indirect association of adverse family environment in childhood with changes of BP over a 10-year period through negative emotions of depression, anxiety or anger.[40] The latter study is not directly comparable to the current study because the questionnaire used only partially addressed verbal and physical abuse and neglect with these assessed as a composite variable.

Additionally, alcohol use partly mediated the relationship between CM and hypertension in the present study. This is in keeping with a systematic review that suggested the association of CM with hypertension may also be influenced by the traditional behavioural risk factors for hypertension.[7] The uptake of risky behaviours was suggested to be coping mechanisms when faced with stressful situations.[5] However, apart from alcohol intake, other traditional risk factors for hypertension such as adiposity, assessed using BMI, did not mediate the associations of CM with hypertension in the RICE study. This may be because the participants in our study were relatively young. Furthermore, our sample was on average overweight-to-obese, restraining the distribution of markers of adiposity in this population. This in turn could affect any reliable investigation of the relationship of adiposity with hypertension in this sample.

### Strengths and limitations
The BP of study participants was objectively measured during data collection, unlike other relevant studies that used a self-reported measure of hypertension

**Table 5** Results of multiple mediation analysis for the effects of childhood maltreatment on adult prevalent hypertension adjusted for age and rape exposure in a sample of South African women (n=1797)

| Independent variables | Total effect | | | Direct effect | | | Indirect effect | | |
|---|---|---|---|---|---|---|---|---|---|
| | Estimates | 95% CI | P value | Estimates | 95% CI | P value | Estimates | 95% CI | P value |
| Any CM | 0.335 | 0.166 to 0.505 | <0.001 | 0.163 | −0.022 to 0.347 | 0.084 | | | |
| BMI | | | | | | | 0.010 | −0.004 to 0.023 | 0.154 |
| HbA1c | | | | | | | −0.002 | −0.015 to 0.011 | 0.768 |
| HIV-positive | | | | | | | 0.002 | −0.008 to 0.011 | 0.694 |
| Current smoking | | | | | | | 0.003 | −0.030 to 0.035 | 0.867 |
| Current alcohol | | | | | | | 0.038 | 0.003 to 0.073 | 0.034 |
| Depressive symptom score | | | | | | | 0.026 | 0.001 to 0.051 | 0.040 |
| ASR score | | | | | | | 0.020 | −0.013 to 0.053 | 0.232 |
| Other traumatic exposures score | | | | | | | 0.076 | 0.040 to 0.113 | <0.001 |
| Multiple CM types | 0.256 | 0.146 to 0.367 | <0.001 | 0.127 | 0.003 to 0.251 | 0.045 | | | |
| BMI | | | | | | | 0.007 | −0.003 to 0.016 | 0.160 |
| HbA1c | | | | | | | −0.001 | −0.009 to 0.007 | 0.872 |
| HIV-positive | | | | | | | 0.003 | −0.004 to 0.011 | 0.414 |
| Current smoking | | | | | | | 0.001 | −0.026 to 0.028 | 0.947 |
| Current alcohol | | | | | | | 0.027 | 0.002 to 0.052 | 0.040 |
| Depressive symptom score | | | | | | | 0.020 | −0.001 to 0.041 | 0.067 |
| ASR score | | | | | | | 0.012 | −0.013 to 0.038 | 0.337 |
| Other traumatic exposures score | | | | | | | 0.061 | 0.032 to 0.090 | <0.001 |
| Cumulative CM score | 0.098 | 0.059 to 0.138 | <0.001 | 0.052 | 0.007 to 0.097 | 0.024 | | | |
| BMI | | | | | | | 0.002 | −0.002 to 0.005 | 0.337 |
| HbA1c | | | | | | | −0.001 | −0.004 to 0.002 | 0.364 |
| HIV-positive | | | | | | | 0.001 | −0.002 to 0.003 | 0.655 |
| Current smoking | | | | | | | 0.000 | −0.010 to 0.010 | 0.976 |
| Current alcohol | | | | | | | 0.009 | 0.001 to 0.018 | 0.038 |
| Depressive symptom score | | | | | | | 0.007 | −0.001 to 0.016 | 0.090 |
| ASR score | | | | | | | 0.004 | −0.006 to 0.015 | 0.425 |
| Other traumatic exposures score | | | | | | | 0.024 | 0.012 to 0.037 | <0.001 |

Data are standardised coefficient; current smoking: currently smoke any tobacco products such as cigarettes, cigars or pipes; current alcohol consumption: consumed at least one drink in an occasion within a month, respectively (one drink is equivalent to one can/bottle of beer, cider, cooler/glass of wine/tot of spirit); depressive symptom (scores): depressive symptoms during the past week using Centre for Epidemiologic Studies Depression Scale (CES-D score); ASR (scores): acute stress reactions measured using Davidson Trauma Scale, a validated self-rating scale. Cumulative abuse (score): ranged from 0 (no abused in any type) to 8 (often/severely abused in all four types).
BMI, body mass index; CM, childhood maltreatment.

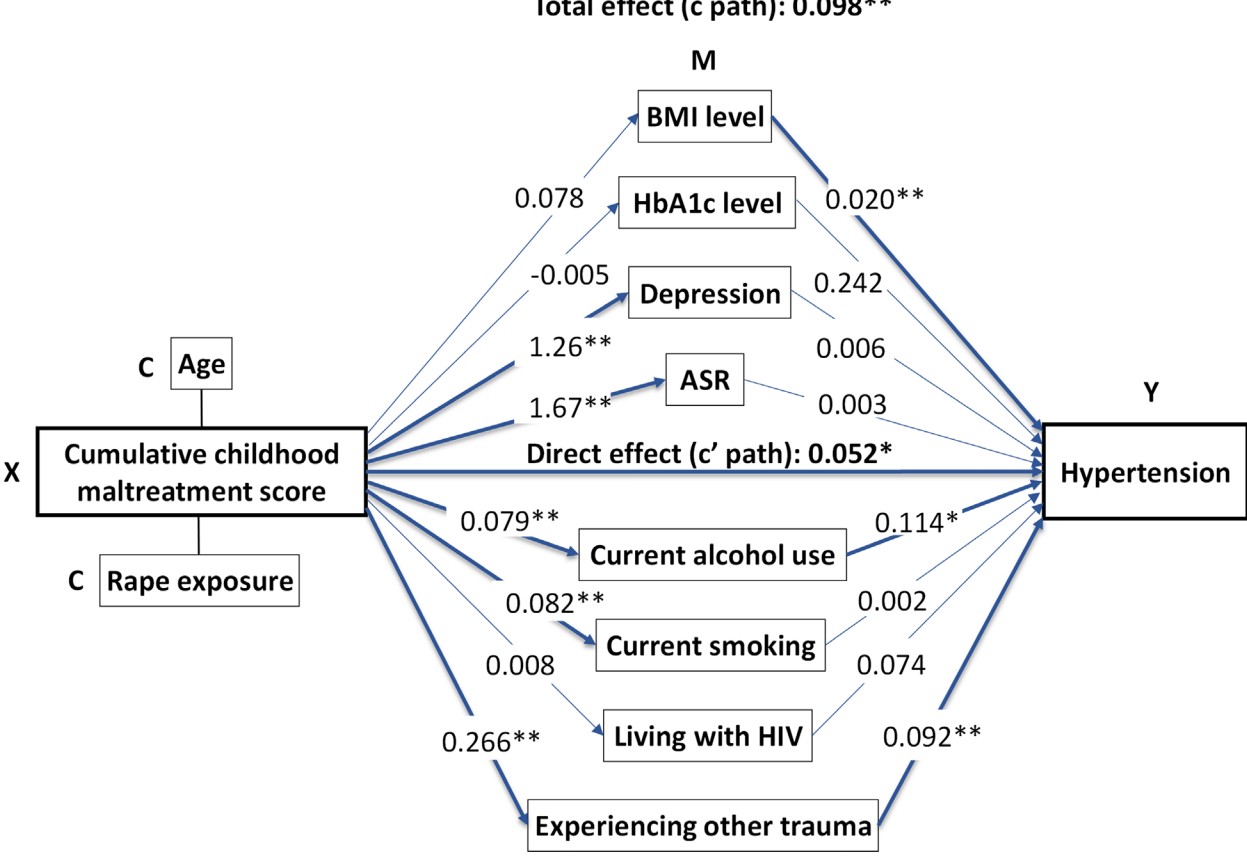

**Figure 1** Multiple mediation model of the association of cumulative childhood maltreatment score with adult prevalent hypertension, adjusted for age and rape exposure. All path estimates are standardised regression coefficients. *p<0.05, **p<0.001. There are significant indirect paths through current alcohol use: 0.009 (0.001–0.018, p=0.038), through experiencing other trauma: 0.024 (0.012–0.037, p<0.001). ASR, acute stress reactions; BMI, body mass index.

only.[36 37][13]This study, therefore, likely captured all participants with hypertension instead of only those who were aware of their hypertension status. Another strength is that this study comprehensively assessed CM by examining four CM types as well as the frequency and severity of the abuse using a well-established locally validated measure.[8] Our study is one of few studies that has assessed the associations of multiple CM types and their frequency and severity with hypertension. This paper, therefore, provides a comprehensive understanding of the relationship between CM and adult hypertension. The present study has several limitations. The cross-sectional design precludes a reliable quantitative interpretation of the effect of the exposure of interest, the mediation effect and the inferences about causality. However, the median age (26 years, 25th–75th: 23–30) of women with hypertension indicates most hypertension onset occurred in adulthood, whereas CM happened in early life. It is likely that this study may have established the temporal relationships between CM and hypertension. It is also known that retrospective self-report of CM experienced could be subject to under-reporting. Although many factors could impact under-reporting of childhood experiences,

suppressed memory of traumatic experiences have been suggested as a key factor.[41]

Furthermore, BP was measured only on a single occasion, which is not a formal diagnosis of hypertension and tends to over-estimate hypertension. Another limitation is that the volunteer sample of women is not representative of all women in the general population. The data were collected only in women and might not be applicable to men. The non-significant influences of BMI on the associations of CM with prevalent hypertension need to be viewed with caution. It could be that our study population was young (median age 24 years, 25th–75th: 21–29), in whom the traditional risk factors may yet influence the development of hypertension. Considering that the RICE study is a longitudinal study, the follow-up data may demonstrate the influence of traditional risk factors on the associations of CM with hypertension later in the life of these women.

## Conclusions

CM is a serious public health problem and associated with adult hypertension independent of traditional risk factors, other traumatic events and mental ill-health in young South African women. The associations of CM with

hypertension were found to be greater in women with multiple maltreatment types and frequent and severe maltreatment, and were partially mediated by alcohol use, current depression, ASR and other traumatic exposures. These findings suggest that prevention of CM is important, not only in itself, but also for the prevention of adult hypertension. Where the prevention of CM is not possible, counselling to manage stress and improve coping skills are important to curb the development of hypertension. This is in line with the goals of psychological therapy for CM survivors which aim to reduce the subsequent health impact. Screening for hypertension in individuals suffering from CM may help the early identification and treatment of the condition. Furthermore, counselling for the other traumatic exposures, and management of depressive symptoms and ASR are also required and may contribute partially to curbing hypertension.

**Contributors** KAN led the analysis and drafted the paper. NP, NA, RJ and APK coconceived and designed the paper. APK, NP, NA, RJ and SM coconceived the broader study, commented on all drafts and approved the final submission. APK assisted with the statistical analysis. KAN and APK are guarantors.

**Funding** The RICE study was funded by South African Medical Research Council: Flagships Awards Project SAMRC-RFA-IFSP-01-2013. The work reported herein was made possible through funding by the South African Medical Research Council through its Division of Research Capacity Development under the SAMRC Postdoctoral Programme from funding received from the South African National Treasury. The content hereof is the sole responsibility of the authors and do not necessarily represent the official views of the SAMRC or the funders.

**Competing interests** None declared.

**Patient and public involvement** Patients and/or the public were not involved in the design, or conduct, or reporting or dissemination plans of this research.

**Patient consent for publication** Consent obtained directly from patient(s).

**Ethics approval** This study involves human participants and was approved by South African Medical Research Council Research Ethics Committee (SAMRC; EC019-10/2013). Participants gave informed consent to participate in the study before taking part.

**Provenance and peer review** Not commissioned; externally peer reviewed.

**Data availability statement** Data are available upon reasonable request. Data are available on request.

**ORCID iD**
Kim Anh Nguyen http://orcid.org/0000-0002-0738-7450

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
