## [Reviewer comments · BMJ Open]

ARTICLE DETAILS

TITLE (PROVISIONAL)	The associations of childhood maltreatment with hypertension in South African women: A cross-sectional study
AUTHORS	NGUYEN, KIM ANH; Kengne, AP; Abrahams, Naeemah; Jewkes, Rachel; Mhlongo, Shibe; Peer, Nasheeta

VERSION 1 – REVIEW

REVIEWER	Devassy, Saju Rajagiri College of Social Sciences, Social Work
REVIEW RETURNED	05-Mar-2022

GENERAL COMMENTS	Comments to the Author The study presents data on the association between childhood maltreatment (CM) with hypertension and the mediating effects of hypertension risk factors on the associations in South African Women. This study addresses a critical and relevant issue of associations of multiple CM types and hypertension. This study results will help to design preventive interventions in future. So, I congratulate the team for choosing such an important area of study. I have a few observations and reservations about this paper. This manuscript is the results of the baseline data of the Rape Impact Cohort Evaluation (RICE) longitudinal study. However, I felt that this single manuscript is stuffed with too much statistical information. Which often resulted in repetition of the information. Further, the authors can correct the grammatical errors. Specific suggestions/comments 1. The sampling procedure/strategy explained in the manuscript requires justification for choosing two groups and pooling the data of two entirely different groups of women. Would it be better to analyse the data separately and compare between the groups as both the groups are qualitatively different in their experiences?2. Theoretical underpinning for the study is not clear even though there are a few vague references found in the discussion section. It would be appreciated if the authors use a strong biopsychological theoretical/ conceptual framework in the background section to establish the rationale for studying the association between CM and hypertension.3. The number of tables can be reduced; e.g., with regard to the second table, presenting the analysis of each question within the subscales does not provide any additional information other than presenting the comparisons of the total score of subscales. The subscales can be computed, and subgroup analysis can be presented to make it simpler. Additionally, since the second and the third table speaks about the same content the authors can omit the second table. The mediating factors are presented in table
--

	form and in the figure. Since the figure is clear enough and sufficient to justify the authors' arguments, the authors can consider omitting that table and adding the most significant information from the table to the figure. 4. Discussion - Authors claimed that "This study confirms CM is highly prevalent..."; Since this is not a prevalence study and the sampling strategy is not suited for prevalence studies the authors' claim will not be acceptable. This study would guide the policymakers and practitioners to incorporate psychosocial components into the hypertension management protocols.
--	--

REVIEWER	Cerna-Turoff, Ilan London School of Hygiene and Tropical Medicine
REVIEW RETURNED	14-Apr-2022

GENERAL COMMENTS	This is an interesting analysis and important in a context like South Africa where there is a high burden of violence and hypertension. Multiple small points and a few larger issues, however, need to be addressed before the manuscript can be published. Larger issues: Mediation analyses  • I am concerned about the mediation analyses conducted as part of this paper. Mediation analysis is best with longitudinal data to untangle temporality. Complete or partial mediation of cross-sectional data tends to be highly biased, and significant mediators in cross-sectional data often are completely different than those in longitudinal data (refer to Maxwell, Cole, and Mitchell 2011 for more discussion). As cross-sectional data, I am not sure that it merits doing mediation analysis at this point to produce likely biased results, especially since you will have longitudinal data to produce more high-quality results in the future. I would remove this analysis entirely. • If you have a good reason to keep the mediation analysis in this paper (which needs to be justified explicitly), some of the mediators—e.g., traumatic experience score over one's lifetime, HIV status depending on time of diagnosis—are not logical, as they could predate child maltreatment. Please ensure that you only select mediators that could actually mediate the relationship between child maltreatment and hypertension. • Less biased methods exist for identifying mediation effects than SEM. I would point you to Imai, Keele, and Yamamoto (2010) and their R package mediation as a potential option. I would also encourage you to do a sensitivity analysis, given the biases that are likely in this data. (2) How was ethnicity/race treated in this study? If you do not adjust for ethnicity or race, it would probably highly bias your results. For instance, BMI as related to hypertension tends to differ by ethnicity so differences in ethnic groups need to be accounted for during the analysis. I would strongly suggest including this as a covariate and rerunning the analyses unless everyone is from the same ethnic/racial group (which should also be mentioned in the manuscript). Smaller suggested changes: General  • The language around trauma is slightly mismatched with the framing used in psychology/mental health literature. There are
--

	several instances (e.g., line 78) where violence and other stressors are themselves described as trauma, rather than individual interpretations and reactions to the events. I would reframe throughout the manuscript. Introduction  • Line 79 which authors warn...? I would cut this sentence entirely since it is unclear and unnecessary for the introduction • Lines 82-83 more clearly “maladaptive and risky behaviours across the lifespan...”) Methods  • You might consider rewording “rape exposed” to “women [or participants] who reported being raped” or something to that extent. • Did women in the control arm self-report not being raped? It would be good to state how this was measured in the manuscript. • Lines 105-111 this section is described in a way that is confusing, because for the purposes of this study, you are pooling the two arms of the study. It is fine to provide background on the treatment and control arms of the study, but then, a roadmap is needed for the reader where you state something along the lines of “In this analysis, we pooled the two arms...”, or alternatively, you could add a couple sentences that introduce the overall methods section. • Interviews were done in 20 days of the rape event – is this a standardized timeframe in which hypertension would be expected to be elevated? Please provide any guidance as to why this timeframe was used and if it is clinically relevant. • Line 118 the wording is a bit confusing. I think you mean that that “an electronic data capture system (Bryant) was used for data collection”. • Line 152 it seems contradictory to say that you cannot determine a PTSD diagnosis and then, say that you sum items to determine a PTSD score. Consider reframing line 152 as a score to measure acute stress reactions. Also, please reframe language about PTSD across the manuscript as measures of acute stress reactions. • For PTSD, depression, and life events, it would be good to mention if these scales were also validated in a South African context, in similar contexts, and/or with this gender or age group. • Line 192 which significance tests were used to determine interactions? Please name which tests were conducted. It also seems like potentially chi-squared significance tests were reported in the results section, but the analysis is not mentioned in the methods section. • Line 202 take out the word causally. You cannot make that claim without much stronger evidence. Discussion  • Line 306 perhaps write as recent poor mental health or something similar since these are not clinical diagnoses. • Lines 321-323 the current study is not examining anxiety, and it is adjusting for self-reported acute stress rather than PTSD. Please amend these two points. I would also remove the word psychological since child maltreatment is not solely mental/verbal. • Line 324 you cannot fully say that the relationship was independent. You can say that an association remained after adjusting for these other factors. • Lines 324-330 a brief summary of this background information on how CM may lead to hypertension seems like it should be placed
--	---

	in the introduction near line 83 to introduce how this may occur physiologically.  • Line 332 please remove additive (you don't know that it is an additive effect) and the word psychological as per above. • Line 343 consider rewording as survivors. • Lines 383-385 this is a big assumption about temporality here. It is enough to state that this is a limitation and leave it at that. You could alternatively add a statistic about the average age of onset for hypertension to say that if maltreatment happened in childhood and most hypertension onset occurs later, you are largely capturing the temporal ordering. • Line 387 underreporting is not just due to suppressed memory. Please revise wording. • Line 389-394 it is confusing here again that you are discussing treatment and control groups when you pooled them for this analysis. You could simply say that the sample of women is not representative of the general population. • A paragraph in the discussion is needed to elaborate further on the possible ramifications of the findings of this study for practice, policy, or the evidence base. Currently, several paragraphs compare the study to the past evidence but do not really highlight what this study adds. Tables  • Table 1 footnote on which kind of significance test was used for p-values • Table 2 footnote on name of scale and age of childhood • Table 4 what is HTN? Please add to footnote. • Table 5 and S1-S3 rape exposure listed as a mediator but then as a covariate in Figure 1 (I believe). Please clarify across tables/figure.
--	--

VERSION 1 – AUTHOR RESPONSE

Reviewer: 1

Dr. Saju Devassy, Rajagiri College of Social Sciences, Rajagiri College of Social Sciences

Comments to the Author:

The study presents data on the association between childhood maltreatment (CM) with hypertension and the mediating effects of hypertension risk factors on the associations in South African Women. This study addresses a critical and relevant issue of associations of multiple CM types and hypertension. This study results will help to design preventive interventions in future. So, I congratulate the team for choosing such an important area of study.

Our answer: We thank you for your appreciation

I have a few observations and reservations about this paper.

This manuscript is the results of the baseline data of the Rape Impact Cohort Evaluation (RICE) longitudinal study. However, I felt that this single manuscript is stuffed with too much statistical information. Which often resulted in repetition of the information. Further, the authors can correct the grammatical errors.

Specific suggestions/comments

1. The sampling procedure/strategy explained in the manuscript requires justification for choosing two groups and pooling the data of two entirely different groups of women. Would it be better to analyse the data separately and compare between the groups as both the groups are qualitatively different in their experiences?

Our answer: Thanks for raising this point. Apart from rape exposure, the two groups by design are comparable for other baseline characteristics, justifying the pooling of the two groups. The approach is further supported by our interactions testing that have shown no evidence of a significant interaction effect of rape exposure on the relationship of CM with hypertension. These details are captured in 1) the Data analyses section on page 8: '*Exploratory analyses revealed no significant interaction effects of 'rape exposure' on the association of CM with hypertension status, motivating the pooled analyses of both rape exposed and unexposed women to maximize the statistical power.*' (Lines 215-219); and 2) the Result section '*There was no evidence of a statistically significant interaction effect of rape exposure on the distribution of CM by hypertension status as shown in Table 3 (all interaction $p \geq 0.061$).*' (Lines 322-324, page 12).

Nonetheless, to account for the effects of recent rape exposure, if any, we adjusted for recent rape exposure in all regression models.

2. Theoretical underpinning for the study is not clear even though there are a few vague references found in the discussion section. It would be appreciated if the authors use a strong biopsychological theoretical/ conceptual framework in the background section to establish the rationale for studying the association between CM and hypertension.

Our answer: Thank you for raising this point and for the suggestion. We have added to the revised manuscript, a paragraph on the postulated hypothesis of biopsychological pathways in the relationship between CM and hypertension. Please see pages 3&4, lines 83-92.

'Growing evidence suggests that CM adversely impacts physical health, resulting in a wide range of disorders including cardiovascular disease risk factors such as hypertension [7,10,11]. It is suggested that exposures to CM may independently activate the hypothalamic-pituitary-adrenal axis and the sympathetic nervous system and promote inflammation, similar to the postulated pathways for other stressors. These physiological changes increase the production of hormones such as catecholamine and glucocorticoid which raise BP levels. The repeated activation of these systems with multiple or frequent exposures to CM may lead to sustained increases in BP and contribute to the development of hypertension [4].'

References:

[4]. Cuevas AG, Williams DR, Albert MA. Psychosocial Factors and Hypertension: A Review of the Literature. *Cardiol Clin* 2017;35:223–30. doi:10.1016/j.ccl.2016.12.004.

[7]. Hughes K, Bellis MA, Hardcastle KA, *et al.* The effect of multiple adverse childhood experiences on health: a systematic review and meta-analysis. *Lancet Public Heal* 2017;2:e356–66. doi:10.1016/S2468-2667(17)30118-4.

[10]. Norman RE, Byambaa M, De R, *et al.* The long-term health consequences of child physical abuse, emotional abuse, and neglect: a systematic review and meta-analysis. *PLoS Med* 2012;9:e1001349. doi:10.1371/journal.pmed.1001349.

[11]. Basu A, McLaughlin KA, Misra S, *et al.* Childhood Maltreatment and Health Impact: The Examples of Cardiovascular Disease and Type 2 Diabetes Mellitus in Adults. *Clin Psychol a Publ Div Clin Psychol Am Psychol Assoc* 2017;24:125–39. doi:10.1111/cpsp.12191

3. The number of tables can be reduced; e.g., with regard to the second table, presenting the analysis

of each question within the subscales does not provide any additional information other than presenting the comparisons of the total score of subscales. The subscales can be computed, and subgroup analysis can be presented to make it simpler. Additionally, since the second and the third table speaks about the same content the authors can omit the second table.

Our answer: We agree with the reviewer, and thus we have moved Table 2 to the online supplementary section for readers interested in those analyses.

3.1. The mediating factors are presented in table form and in the figure. Since the figure is clear enough and sufficient to justify the authors' arguments, the authors can consider omitting that table and adding the most significant information from the table to the figure.

Our answer: Thank you for the suggestion. In Table 5, we presented the results of simple mediation analysis, which we still believe is the optimal approach. In fact, we have 3 exposures (any CM, multiple CM, and cumulative CM), with each having eight mediators which were modelled separately; and thus, would result in 24 figures, if we were to go only with figures. Such a number is likely too high for the manuscript.

Figure 1, which presents the results of the multiple mediation model for the effect of cumulative CM is mostly for demonstration purpose, while the full results are shown in Table S2. We have now edited the Figure 1 to make it more informative, as suggested by the reviewer.

4. Discussion - Authors claimed that "This study confirms CM is highly prevalent..."; Since this is not a prevalence study and the sampling strategy is not suited for prevalence studies the authors' claim will not be acceptable.

Our answer: Thank you for raising this point. We have changed the wording to 'widespread'. The relevant sentence on page 13 (lines 369-370) has been modified to read: "In this study among South African adult women attending specific healthcare services, CM was widespread and associated with hypertension."

This study would guide the policymakers and practitioners to incorporate psychosocial components into the hypertension management protocols.

Our answer: Thank you.

Reviewer: 2

Dr. Ilan Cerna-Turoff, London School of Hygiene and Tropical Medicine

Comments to the Author:

This is an interesting analysis and important in a context like South Africa where there is a high burden of violence and hypertension. Multiple small points and a few larger issues, however, need to be addressed before the manuscript can be published.

Our answer: Thanks for your appreciation

Larger issues:

Mediation analyses

- I am concerned about the mediation analyses conducted as part of this paper. Mediation analysis is best with longitudinal data to untangle temporality. Complete or partial mediation of cross-sectional data tends to be highly biased, and significant mediators in cross-sectional data often are completely different than those in longitudinal data (refer to Maxwell, Cole, and Mitchell 2011 for more discussion). As cross-sectional data, I am not sure that it merits doing mediation analysis at this point

to produce likely biased results, especially since you will have longitudinal data to produce more high-quality results in the future. I would remove this analysis entirely.

Our answer: We agree with the reviewer's view that mediation analysis is best with longitudinal data to untangle temporality. The mediation analysis in cross-sectional studies is subject to some bias due to the nature of the cross-sectional design. The mediators were based on assumptions, and thus the key limitation is that it cannot be used for inference about causality. Accordingly, we acknowledged this limitation on page 16, lines 450-452, where it reads: 'The cross-sectional design precludes a reliable quantitative interpretation of the effect of the exposure of interest, the mediation effect and the inferences about causality.'

However, with the paucity of data from longitudinal studies in most settings, recent literature shows that the mediation analysis of cross-sectional studies has gained popularity, especially in the field of psychology and psychiatry research to determine possible links between exposures and mental and physical health outcomes. There have been several publications applying mediation analyses with a cross-sectional data, prior to our study (refs showed below). Furthermore, like regression analyses in cross-sectional design, the mediation analysis in cross-sectional studies will give an idea of where longitudinal studies as well as interventions should be focused on. For the current study, childhood maltreatment occurred in early life, and prevalent hypertension is current, and so this study set some sort of temporality between CM and hypertension. However, one cannot affirm with certainty that all the mediators occurred between the exposure to CM and the subsequent onset of hypertension. Despite the fact that the RICE study is longitudinal, the short follow-up time and the young age of the population would translate into fewer incident hypertension for reliable mediation analyses using longitudinal data.

References:

Watt MH, Ranby KW, Meade CS, *et al.* Posttraumatic stress disorder symptoms mediate the relationship between traumatic experiences and drinking behavior among women attending alcohol-serving venues in a South African township. *J Stud Alcohol Drugs* 2012;73:549–58.

doi:10.15288/jsad.2012.73.549

Machisa MT, Christofides N, Jewkes R. Structural Pathways between Child Abuse, Poor Mental Health Outcomes and Male-Perpetrated Intimate Partner Violence (IPV). *PLoS One* 2016;11:e0150986. doi:10.1371/journal.pone.0150986

doi:10.1371/journal.pone.0150986

Wright EN, Hanlon A, Lozano A, *et al.* The impact of intimate partner violence, depressive symptoms, alcohol dependence, and perceived stress on 30-year cardiovascular disease risk among young adult women: A multiple mediation analysis. *Prev Med (Baltim)* 2019;121:47–54.

doi:10.1016/j.ypmed.2019.01.016

- If you have a good reason to keep the mediation analysis in this paper (which needs to be justified explicitly), some of the mediators—e.g., traumatic experience score over one's lifetime, HIV status depending on time of diagnosis—are not logical, as they could predate child maltreatment. Please ensure that you only select mediators that could actually mediate the relationship between child maltreatment and hypertension.

Our answer: In this study, we sought to investigate the associations of CM with prevalent hypertension, and to explore the potential mediating role of conventional risk factors for hypertension. We have now added the following explanation to the Methods section on page 9, lines 234-248.

'The potential mediators were based on the existing literature and included BMI, HbA1c, smoking, alcohol drinking, HIV infection, depressive symptoms, ASR and other traumatic event experiences. They were postulated to be involved in the pathways linking CM with hypertension. HIV infection was considered as a potential mediator because previous studies in South Africa have shown CM to be associated with HIV-related risk behaviour (Gibbs *et al.*), and living with

HIV infection to be associated with higher risk for hypertension (Davis 2021). On rare occasions, HIV infection could have been acquired before CM exposure, but the number would likely be too few to bias the mediation analyses. Other traumatic event experience was positioned as a potential mediator based on the local literature (Gass et al 2011). Although the exact order of occurrence between CM and other traumatic event exposure could not be specified in the current study, CM, by definition, typically occurred only in childhood. Some traumatic events could have happened at any time during the lifespan, and have been conventionally measured through scores that capture the lifetime experience, which were used in the current study. Lifetime experience of these other traumatic events could have been the consequence of CM or not; but CM would very unlikely be the consequence of other traumatic event exposure.'

References:

Gibbs A, Dunkle K, Washington L, *et al.* Childhood traumas as a risk factor for HIV-risk behaviours amongst young women and men living in urban informal settlements in South Africa: A cross-sectional study. *PLoS One* 2018;13:e0195369. doi:10.1371/journal.pone.0195369.

Davis K, Perez-Guzman P, Hoyer A, *et al.* Association between HIV infection and hypertension: a global systematic review and meta-analysis of cross-sectional studies. *BMC Med* 2021;19:105. doi:10.1186/s12916-021-01978-7.

Gass JD, Stein DJ, Williams DR, *et al.* Intimate partner violence, health behaviours, and chronic physical illness among South African women. *S Afr Med J* 2010;100:582–5. doi:10.7196/samj.4274

• Less biased methods exist for identifying mediation effects than SEM. I would point you to Imai, Keele, and Yamamoto (2010) and their R package mediation as a potential option. I would also encourage you to do a sensitivity analysis, given the biases that are likely in this data.

Our answer: Thanks for this suggestion. We have replicated both simple and multiple mediation analyses using 'mediation' package and presented the findings as secondary analyses in Supplementary Tables S4 and S5. Patterns were mostly similar to those from the SEM analyses, and have now been indicated in the result section. Please see the revised manuscript, page 13, lines 352-367.

(2) How was ethnicity/race treated in this study? If you do not adjust for ethnicity or race, it would probably highly bias your results. For instance, BMI as related to hypertension tends to differ by ethnicity so differences in ethnic groups need to be accounted for during the analysis. I would strongly suggest including this as a covariate and rerunning the analyses unless everyone is from the same ethnic/racial group (which should also be mentioned in the manuscript).

Our answer: Thank you. Most women in the RICE study were black (> 95%) and no further details on ethnicity were collected. We have now indicated in the manuscript that 'our participants are mostly black women ...'. It is of note that data collection on Race/ethnicity in studies in South Africa is a very sensitive issue which Ethics Committees are extremely cautious about. Accordingly collecting these data is generally discouraged.

Smaller suggested changes:

General

• The language around trauma is slightly mismatched with the framing used in psychology/mental health literature. There are several instances (e.g., line 78) where violence and other stressors are themselves described as trauma, rather than individual interpretations and reactions to the events. I would reframe throughout the manuscript.

Our answer: We have now replaced 'psychological' with 'mental' or 'mental health' where appropriate throughout the manuscript.

Introduction

- Line 79 which authors warn...? I would cut this sentence entirely since it is unclear and unnecessary for the introduction.

Our answer: Thank you. Suggestion effected.

- Lines 82-83 more clearly “maladaptive and risky behaviours across the lifespan...”)

Our answer: We have amended the sentence to read as: ‘The impact of CM is well-established with CM-induced traumatic stress contributing to depression, anxiety, post-traumatic stress and maladaptive risky lifestyle behaviours in later life’

Methods

- You might consider rewording “rape exposed” to “women [or participants] who reported being raped” or something to that extent.

Our answer: Thank you. We have reworded accordingly.

‘The study included women who reported being raped (rape exposed) (n=947) recruited from five public health-based post-rape care service and women who have not ever experienced rape or forced sex (non-rape exposed)’. Page 5, lines 114-117.

- Did women in the control arm self-report not being raped? It would be good to state how this was measured in the manuscript.

Our answer: The inclusion criteria for the control group was not ever having experienced a rape. We indicated this in the manuscript: ‘Further exclusion among women in the control arm were those who reported lifetime exposure to rape or force sex.’. Page 5, lines 123-124.

- Lines 105-111 this section is described in a way that is confusing, because for the purposes of this study, you are pooling the two arms of the study. It is fine to provide background on the treatment and control arms of the study, but then, a roadmap is needed for the reader where you state something along the lines of “In this analysis, we pooled the two arms...”, or alternatively, you could add a couple sentences that introduce the overall methods section.

Our answer: Thanks for raising this point and for the recommendation. We have added the following to the manuscript:

In the overall Method section:

‘Although the RICE study has two arms, in an attempt to achieve the desired aims, the present study used the data from the pooled group of women, that are described more detail in the statistical analysis.’ Page 5, lines 129-131.

In the statistical analysis:

‘Since this study aimed to determine the associations of CM with adult hypertension in both rape-exposed and non-rape-exposed women, and not to determine and/or compare the prevalence of CM and outcome hypertension between the two groups of women...’ Page 8, lines 211-214.

- Interviews were done in 20 days of the rape event – is this a standardized timeframe in which hypertension would be expected to be elevated? Please provide any guidance as to why this timeframe was used and if it is clinically relevant.

Our answer: The baseline interview for the study was done within 20 days of the rape event. This is not standardise practice but a critical aspect of the main study. The decision to do the baseline interview close to the index rape was in response to the primary aim of the main study which was to

determine the incidence of HIV infection after rape and to determine the mental health status. This initial interview allowed for the baseline data on HIV status and documentation of acute stress reactions. This is also explained in the protocol publication which is references on page 5 line 121. We have amended this section to explain this better. Page 5 Lines 126-128.

- Line 118 the wording is a bit confusing. I think you mean that that “an electronic data capture system (Bryant) was used for data collection”.

Our answer: Yes. Thank you. We have changed the sentence accordingly. Please see page 5, line 133.

- Line 152 it seems contradictory to say that you cannot determine a PTSD diagnosis and then, say that you sum items to determine a PTSD score. Consider reframing line 152 as a score to measure acute stress reactions. Also, please reframe language about PTSD across the manuscript as measures of acute stress reactions.

Our answer: Thank you. We have changed PTSD to ASR (acute stress reactions) throughout.

- For PTSD, depression, and life events, it would be good to mention if these scales were also validated in a South African context, in similar contexts, and/or with this gender or age group.

Our answer: The ASR, depression, and life events questionnaires were validated in many international settings and in South Africa. We had incorporated this information in the paper on Pages 6- 7. Lines 168,169; 174,175, 181.

- Line 192 which significance tests were used to determine interactions? Please name which tests were conducted. It also seems like potentially chi-squared significance tests were reported in the results section, but the analysis is not mentioned in the methods section.

Our answer: We have added to the Statistical analysis section the following statements to clarify these points:

‘The interactions were tested in logistic regression models which included the interaction term (CM*Rape-exposure) together with the main effects of CM and rape-exposure as predictors, and prevalent hypertension as the outcome.’ Page 8, lines 219-221.

‘Comparison of baseline characteristics by hypertension status used chi-square tests, fisher-exact test, t-tests or Kruskal-Wallis tests as appropriate.’ Page 8, lines 224-226.

- Line 202 take out the word causally. You cannot make that claim without much stronger evidence.

Our answer: Done.

Discussion

- Line 306 perhaps write as recent poor mental health or something similar since these are not clinical diagnoses.

Our answer: Done. We have changed from ‘mental health illness to ‘recent poor mental health’. Line 374, the revised paper.

- Lines 321-323 the current study is not examining anxiety, and it is adjusting for self-reported acute stress rather than PTSD. Please amend these two points. I would also remove the word psychological since child maltreatment is not solely mental/verbal.

Our answer: We amended the paragraph accordingly. The paragraph now read as 'However, these two studies, unlike the current study, did not investigate the effects of mental health issues such as depression and ASR, on the relationship between CM and hypertension. Notably, the current study found that the associations of CM with hypertension remained after adjusting for poor mental health-related factors'. Page 13, lines 378, 389-393.

- Line 324 you cannot fully say that the relationship was independent. You can say that an association remained after adjusting for these other factors.

Our answer: We have amended accordingly. Please see our answer for the above point.

- Lines 324-330 a brief summary of this background information on how CM may lead to hypertension seems like it should be placed in the introduction near line 83 to introduce how this may occur physiologically.

Our answer: Thank you. We have moved the paragraph to the Introduction, page 4, lines 86-92.

- Line 332 please remove additive (you don't know that it is an additive effect) and the word psychological as per above.

Our answer: Suggestions effected. We have replaced the word 'psychological' with 'mental health-related'.

- Line 343 consider rewording as survivors.

Our answer: Suggestion effected.

- Lines 383-385 this is a big assumption about temporality here. It is enough to state that this is a limitation and leave it at that. You could alternatively add a statistic about the average age of onset for hypertension to say that if maltreatment happened in childhood and most hypertension onset occurs later, you are largely capturing the temporal ordering.

Our answer: Thank you for this advice. We have amended the paragraph to read as:

'However, the median age (26 years, 25th-75th: 23-30) of women with hypertension indicates most hypertension onset occurred in adulthood, whereas CM happened in early life. It is likely that this study may have established the temporal relationships between CM and hypertension. Page 16, lines 452-454.

- Line 387 underreporting is not just due to suppressed memory. Please revise wording.

Thank you for pointing this out. We have changed the wording to explain this better. See page 16 lines 457-459. This is changed to the following, 'Although many factors could impact underreporting of childhood experiences, suppressed memory of traumatic experiences have been suggested as a key factor.'

- Line 389-394 it is confusing here again that you are discussing treatment and control groups when you pooled them for this analysis. You could simply say that the sample of women is not representative of the general population.

Our answer: Thank you. We have now removed 3 sentences referred to two groups of women, and added the following 'Another limitation is that the volunteers sample of women is not representative of all women in general population.' Page 16, lines 461-462.

- A paragraph in the discussion is needed to elaborate further on the possible ramifications of the findings of this study for practice, policy, or the evidence base. Currently, several paragraphs compare the study to the past evidence but do not really highlight what this study adds.

Our answer: In the Conclusion section, we now state the following:

'These findings suggest that prevention of CM is important, not only in itself, but also for the prevention of adult hypertension. Where the prevention of CM is not possible, counselling to manage stress and improve coping skills are important to curb the development of hypertension. This is in line with the goals of psychological therapy for CM survivors which aim to reduce the subsequent health impact. Screening for hypertension in individuals suffering from CM may help the early identification and treatment of the condition. Further, counselling for the other traumatic exposures, and management of depressive symptoms and ASR are also required and may contribute partially to curbing hypertension.' Pages 17, lines 480-488.

Tables

- Table 1 footnote on which kind of significance test was used for p-values

Our answer: Done.

- Table 2 footnote on name of scale and age of childhood

Our answer: We amended accordingly. Please note that we have now moved Table 2 to the online supplement.

- Table 4 what is HTN? Please add to footnote

Our answer: Done!

- Table 5 and S1-S3 rape exposure listed as a mediator but then as a covariate in Figure 1 (I believe). Please clarify across tables/figure.

Our answer: We have amended in the Method section to clarify this point.

Reviewer: 1

Competing interests of Reviewer: Nil

Reviewer: 2

Competing interests of Reviewer: I have no competing interests

VERSION 2 – REVIEW

REVIEWER	Devassy, Saju Rajagiri College of Social Sciences, Social Work
REVIEW RETURNED	21-Jul-2022
GENERAL COMMENTS	I am satisfied with the changes made and I think it can be published in the current form.